# Assessment of the Water-Energy Nexus under Future Climate Change in the Nile River Basin

**Abay Yimere [1,2,\*] and Engdawork Assefa [1]**

1   College of Development Studies, Center for Environment and Development, Addis Ababa University, Addis Ababa P.O. Box 9086, Ethiopia; engdawork.assefa@aau.edu.et

2   Research Affiliate, Fletcher School of International Law and Diplomacy, Tufts University, 160 Packard Ave, Medford, MA 02155, USA

\*   Correspondence: abay.ezra@aau.edu.et; Tel.: +251-9441324445

**Abstract:** This study investigated the Water-Energy relationship in the Nile River Basin under changing climate conditions using an energy and water model. Climate change will likely affect both water and energy resources, which will create challenges for future planning and decision making, particularly considering the uncertainty surrounding the direction and magnitude of such effects. According to the assessment model, when countries depend heavily on hydropower for energy, power generation is determined by climate variability. For example, Ethiopia, Egypt, and Sudan are more hydropower-dependent than Burundi or Rwanda. As a result, the trading relationships and economic gains of these countries shift according to climate variability. Among 18 climate scenarios, four demonstrate a change in climate and runoff. Under these scenarios, trading partnerships and economic gains will favor Ethiopia and Egypt instead of Sudan and Egypt. This study examines the extent of potential climate challenges, their effects on the Nile River Basin, and recommends several solutions for environmental planners and decision makers. Although the proposed model has the novel ability of conducting scientific analyses with limited data, this research is still limited by data accessibility. Finally, the study will contribute to the literature on the climate chamber effects on regional and international trade.

**Keywords:** water model; energy model; climate scenario; Nile River Basin

## 1. Introduction

Water-Energy models are regularly updated to incorporate new developments; thus, they often vary in their technological requirements, data specifications, and computing capabilities. Many Water-Energy models cannot be applied to developing countries because data, high-computational demands, and skill requirements cannot be used to their full extent. This is because most models are developed in industrial countries and designed to match their technological development. Moreover, most energy models are not appropriate for developing countries because of the models' requirements, functions, and objectives [1,2]. For example, Nakata [1] examined models related to the energy environment, and Pandey [2] highlighted the importance of integrating the unique features of developing countries into the design and development of Water-Energy models. Furthermore, the data gap is a critical challenge for developing countries because it limits scenario discovery, hinders technological advances, and derails the impacts of policy analysis [3].

In light of these challenges, we developed a model to assess the Water-Energy relationship in the Nile River Basin that captures its unique riparian characteristics and conditions, including the informal energy sector, income and consumption, centralized market and supply options, and changing temporal patterns. As well as an integrative approach, the proposed accounting framework also emphasizes scenario discovery with limited data and fosters a data exchange between models until convergence is reached. Thus, the aim of this

study is to apply this tailored Water-Energy model to the Nile River Basin to investigate the regional effect of climate change on water and energy resources.

*Literature Review*

Strong links have long been recognized between water-resource and energy systems because water is essential to energy production. As of 2016, the energy sector accounted for approximately 15% of all fresh water use worldwide [4]. For the majority of developing nations in Africa, where hydropower is the principal source of energy, water and energy cannot be viewed separately. However, decision makers and policy makers had, until recently, overlooked the strong interdependence of these sectors, which has often led to situations where resources were either underused or exploited in a non-sustainable manner [5]. Thus, with growing demands for water in the food and industry sectors, as well as ever-increasing energy demands, an integrated assessment of the Water-Energy nexus has become even more relevant when evaluating alternatives for better decision making and management.

Climate change will likely affect both water and energy resources, which will create challenges for future planning, particularly considering the uncertainty surrounding the direction and magnitude of such effects. Rising temperatures will likely lead to an increased demand for irrigation due to increased evapotranspiration [6]. This additional need to meet the potential evapotranspiration (PET) demand or address additional evaporation losses will affect water allocation, with hydropower particularly vulnerable to a drier future [7]. Thus, a more inclusive analysis is required to evaluate future water and energy risks under a changing climate and assess the resiliency of a given Water-Energy system to resource variability and other competing demands.

Energy system models are important tools for analyzing future energy supply and demand at the national, state, and regional level under certain assumptions, such as developmental scenarios, electricity prices, availability, and energy-generation capacity [8]. Historically, energy accounting has been one of the key pillars of energy system studies, as it provides insight into the overall balance of an energy system [9]. Accordingly, Hoffman and Wood [10] recommended the energy-accounting approach as an essential framework for energy system research. Long-range Energy Alternative Planning (LEAP) is an extension of this approach, which addresses the recommendations of subsequent studies. However, the reference energy system (RES) is a natural outcome of the energy balance system [11]. RES audits all existing events across an energy supply chain by considering its technological level, scope, and features. The method developed by Hoffman and Wood [10] helps incorporate current and future technological options to enable the system to perform analyses. Following RES development and calibration, linear programming was developed to expand further and integrate the model. Subsequently, a model known as Brookhaven Energy System Optimization (BESOM) was developed for the purpose of resource allocation [12]. After linear programming and the development of BESOM, various models were developed and human abilities were increased, enabling the analysis of economic linkages through input–output analysis at national and regional levels. Historically, BESOM has served as a basis for many other model developments and derivatives [13].

Energy models have been categorized into three types based on the model approach: top-down, bottom-up, and hybrid [14–19]. Each type of model has a different aim. The bottom-up approach begins by describing the technologies for supply and demand; top-down models start by explaining the relationship among several components at the macroeconomic level; and hybrid approaches attempt to combine features of these two models. Various efforts have been made to integrate the bottom-up and top-down approaches into a hybrid model [16,17]. Typically, the top-down approach is more applicable to econometric, input–output, and general equilibrium features. The programing techniques (linear, nonlinear, mixed-integer, and neural theories) also describe the top-down approach. Conversely, bottom-up models are more applicable to optimization and simulation. Previous studies have also attempted to categorize models based on functionality [16,17].

Energy models have also been reviewed and evaluated based on their scope: from individual projects to multifaceted and global systems and from long-term to standalone projects [20–22]. Long-term models also pursue either a top-down, bottom-up, or hybrid approach. However, long-term models must adapt to long-term changes and have their parameters updated accordingly [17,23–25]. To that end, the long-term model embraces a system-wide approach. Major long-term bottom-up models include the Energy Flow Optimization Model (EFOM) and MARKAL [26]. EFOM, which was developed under the authorization of the European Commission, is a bottom-up engineering-oriented model designed to support regional energy strategies and policies [27]. MARKAL is, along with TIMES, a successful derivative of BESOM used by the European Commission to simulate the energy–environment system at the global, European, national, and community levels. The input data can easily tailor the model, which can capture energy supply-and-demand evolution for up to 100 years [28,29]. MESSAGE, which was later enhanced into MESSAGE I and MESSAGE II, is also considered to be an early generation of an optimization model. Previously, the World Energy Council and the Intergovernmental Panel on Climate Change used MESSAGE to develop energy transition pathways and greenhouse gas emission scenarios, respectively [30,31]. The Open-Source Energy Modeling System (OSeMOSYS) also belongs to the class of early bottom-up models [32].

In the 1990s, bottom-up models flourished, particularly with development of the Prospective Outlook on Long-term Energy Systems (POLES), World Energy Model (WEM), PRIMES, and LEAP. PRIMES can be modular [33] and is strongly linked to the generation of the PROMETHEUS mode. LEAP is essentially a simulation-model framework with minor input data requirements, and combines a top-down demand with a bottom-up supply [22,34]. AURORAxmp [35], EPI [36], CYME EATON [37], DER-CAM Distributed Energy Resources Customer Adoption Model [38], EMPS [39], Enertile [40], ENTIGRIS (Energy System Models at Fraunhofer ISE0 [41], ETSAP-TIAM [26], and PLEXOS (Energy Exemplar) [42] are all categorized as bottom-up models. PLEXOS, which was later extended to MOSEK and Xpress-MP, was developed as a linear and mixed-integer model. PLEXOS develops scenarios to capture regional markers and prices to perform market design and analysis with hydrological, thermal, and transmission features. AURORAxmp was designed to examine hydropower generation and load volatility under uncertain conditions. Likewise, in the 1980s, an expanded version of a top-down model was developed, called Phoenix, an extension of the general equilibrium model (GEM). Following its development, top-down models became prevalent, including the General Equilibrium Environmental Model (GREEN), MARKAL, GEM-E3, and EPPA [43–49]. Under the approval of the Secretariat of the Organization for Economic Cooperation and Development (OECD), GREEN has become a global model to evaluate the impact of economic activities on abating $CO_2$ emissions [50]. MARKAL was adapted to a top-down model to create both MARKAL–MACRO [51] and MARKAL–EPPA [45]. Hybrid energy models also include RETScreen [52], Natural Resources Canada [53], POLES [54], MESSAGE [55], LEAP [56], GCAM (Joint Global Change Research Institute) [57], and ETM [58].

In dealing with uncertainties, a stochastic approach is more desirable compared to a deterministic approach, which must be calibrated several times by changing input similar to that of Monte Carlo. This approach works like MARKAL's version of stochastic [59] and MESSAGE's stochastic [60]. Rigorous stochastic models for Water-Energy nexus systems have recently been developed, and they are considered vital for their meticulous results [61]. Concerning deterministic model studies, the supply side of the water nexus gets less attention given its optimization uncertainties [62]. One model was employed in a study of a transboundary water nexus (Albrecht et al., 2018) and more recently one was used in an analysis of the nexus in the Mekong river basin [63].

Some previous energy models have been developed and refined from case studies and experiments. For example, Welsch [20] further refined the open-source OSeMOSYS toolkit and TIMES—PLEXOS model to investigate the Irish energy system; Poncelet [64] employed TIMES to investigate the Belgium energy system; Haydt [65] also employed

LEAP, MARKAL/TIMES, and EnergyPLAN to investigate the Atlantic Ocean energy system in the Atlantic Ocean for the Azores; and Jaehnert and Doorman [66] combined EMPS and IRIE to understand the energy balance between input-out energy systems.

Furthermore, numerous tools have been suggested to improve decision making related to water resource management. For example, Mysiak et al. [67] proposed the integration of hydrological tools, and Rees et al. [68] developed a water balance model that calculates the balance between water supply and demand. Moreover, Li et al. [69] proposed a water resource management model that, under uncertain conditions, is expected to help decision makers identify and develop a response system for uncertain resource challenges. Another model developed by Van Cauwenbergh et al. [70] prioritizes water resource planning and management according to the requirements of environment and socioeconomic development. A fuzzy-set mathematical theory has also been proposed and recommended by researchers to address water resource management challenges through a robust decision-making process [71,72]. The Water Evaluation and Planning (WEAP) model developed by the Stockholm Environmental Institute (SEI) has been used to evaluate various water resource planning and management alternatives [73,74]. WEAP allocates water according to user-developed criteria, and the primary use of this model is for scenario development, through which it answers various "what if" questions pertaining to the demand and supply of water [75,76].

Most importantly, the stochastic models have shown superiority in providing long-term analysis of uncertainty and in helping grasp natural behaviors [77]. Harold Edwin Hurst, who studied the Nile River for about 60 years, observed that the time series of the river's annual flow displays statistical behaviors that do not fit into a series of simply random variation; instead, their tendency to occur in natural events are higher, of which these characteristics are known as Hurst Phenomena or Long–term Persistence (LTP) [78,79]. Scientists have used these extended time horizon research methods to study ensembles of synthetic hydrologic conditions that take annual, seasonal and decadal variability into consideration, including studies of the Boeoticos Kephisos Basin in Greece [80], the Nile River Basin in Africa [81], and documented the presence of LTP in precipitation [82], runoff [83], temperature [84] and in the hydrological cycle [85].

In conclusion, different energy models are applied according to their approach and techniques. Among the bottom-up energy models, OSeMOSYS [20,86] the Integrated MARKAL–EFOM System (TIMES) [26,27], and LEAP [21,74] have all been widely applied. Equally, water planners commonly employ the WEAP system [48,51,75], and RiverWare [87,88] as water system analysis modeling tools. In most cases, these two categories of models are applied separately using the energy model's output as the input to the water model and vice versa. Presently, the development of comprehensive models that consider both energy and water and model their systemic interactions is very limited. Of the few studies that have demonstrated this interaction, one describes an integrated approach using a water system in WEAP and an energy model in LEAP in Sacramento, California, [89] and one that took a climate, land use, energy, and water systems (CLEWs) approach, which involved a more comprehensive module-based approach where data were exchanged between the sectoral models in an iterative fashion [90].

Most existing energy system models are in an early stage of integrating energy and water, and lack temporal representation. Therefore, this study presents an integrated water and energy system model that captures the links between these sectors. A regional power integration energy model and a basin-wide water resource system model are coupled and simulated together to investigate how future climate change might impact the generating capacity of existing and future planned hydropower plants in the Nile River Basin, as well as the dynamics of trade between plants in the region. The main feature of this framework is the use of an iterative approach to exchange data between the models until convergence is established in the linked components. The model is applied to regional trade in the countries of the Nile River Basin under the Eastern Africa Power Pool (EAPP) and the future development scenario of the Program for Infrastructure Development in Africa

(PIDA) [91]. The impact of climate change is categorized and evaluated for scenarios of an unconstrained scenario, with unrestricted emissions of greenhouse gases and a future scenario, in which a restriction policy is imposed to limit emissions to a certain level, referred to as level one stabilization. These two scenarios reveal the possible advantages that could be achieved under a policy of adaptation.

## 2. Methods

A key element of the coupled Water-Energy modeling approach presented in this study involves simulating both the water system (to estimate hydropower generation constrained by water availability) and the energy system, applying feedback loops between the two models until equilibrium is achieved between hydropower generated from the water model and hydroelectric energy used in the energy model on a national level. To analyze the impact of climate change, the hydrologic model was simulated to estimate runoff corresponding to different climate scenarios (characterized by different changes in temperature and precipitation) to establish water availability for the water system model. Then, the coupled Water-Energy system was simulated for each climate change scenario. This configuration considers energy demands according to both water availability and energy cost after the energy mix is identified.

### 2.1. Energy and Water System Models

### 2.1.1. Energy Model

The energy system was modeled using the Regional Integration and Planning Assessment (RIPA) tool [8,24], configured in the General Algebraic Modeling System (GAMS) mathematical programming language [92,93]. RIPA is a bottom-up, dynamic, multiyear optimization program that makes use of mixed-integer programming (MIP) techniques to solve the optimal mix of generation and transmission by minimizing all discounted investment and operating costs while meeting the demands for different specified energy sources over the planning period. One of the inputs of RIPA is the generation technology and corresponding monthly available capacity over the simulation period. Monthly hydropower generation is one of the renewable energy inputs to the model that serves as the upper boundary of the maximum available resource from hydropower when solving for the country-level energy balance. After the optimal mix of generation is solved, the amount of energy not utilized in any of the technologies is reported as a "spill", together with the shadow prices obtained for the bounding constraints in the MIP optimization.

### 2.1.2. Water Model

The water resource system model simulates water allocation and hydropower generation from available resources by solving different Linear Programming (LP) problems that are defined iteratively at each time step in monthly intervals [94,95]. These problems are determined based on the priorities and nature of water and power demand and stream flow requirements. Hydropower generation is calculated from the flow passing through the turbine having a maximum capacity to fulfill the specified monthly energy demand. Hydropower generation is constrained by the available water in the system and the priority assigned to the demand. The model reports the total energy generation from each plant and the power deficit, i.e., the difference between the specified monthly energy demand and generation.

### 2.2. Formulation of Model Coupling

Although water and energy production/use can be interlinked in several ways, this study explored only hydropower generation and demand. The primary objective was to minimize spillage from the energy model and shift excess energy to other months in the same year by storing water. Initially, hydropower estimation was based on estimates of the country-level target demand. For storage hydropower plants, the initial estimates were taken from the average generation, whereas for existing hydropower plants, a rough

approximation was calculated as a percentage of the planned installed capacity for future infrastructure. For run-of-river hydropower plants, the maximum generating capacity was taken as the power target for each month. As their storage capacity is small or nonexistent, they are configured for maximum potential generation based on the available water in the system. However, the generation capacity for storage reservoirs can be constrained by water availability, priority demands for water that are higher than the level of energy generation, and reservoir rule curves used in addition to the specified demand. The information exchange between the water and energy system models and an example of model interaction are shown in Figure 1.

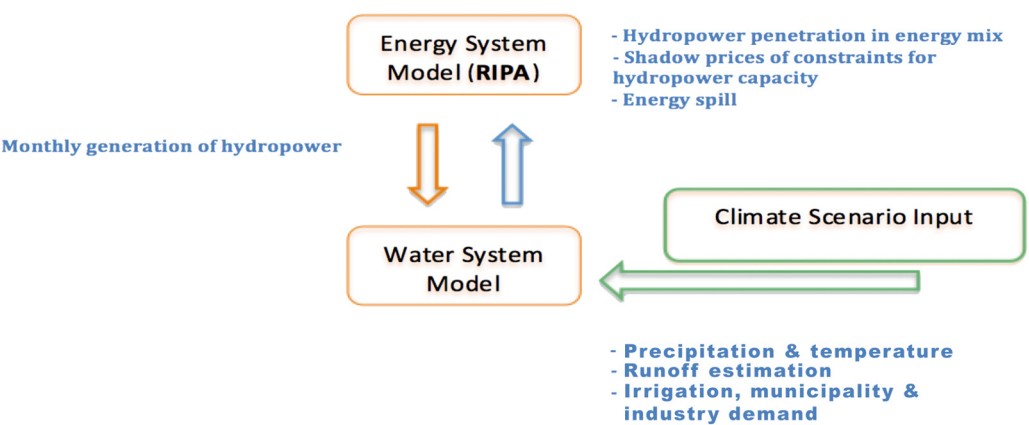

**Figure 1.** Schematic showing the Water-Energy interaction and climate modules. RIPA = Regional Integration and Planning Assessment tool.

An initial simulation of the water system was conducted to estimate the monthly energy demand at the plant level, which produces a time series of the monthly generated power. This power generation is then aggregated by country and used as an input in the energy system model to determine the optimum mix of energy for each sector and the unused energy "spill". From the energy mix, the hydropower outputs are then disaggregated back to the plant-level generation based on the ratio of the total generated energy determined from the water model to the energy demand determined for each plant, which is used for the next iteration, given in Equation (1). This ensures that the amount of generation from the water model is fully used in the energy mix.

$$Energy\ Demand_{plant,year} = HydroShare_{country} * \frac{Hydro\ Generation_{plant}}{\sum_{country} Hydropower\ Share\ in\ Energy\ mix} \qquad (1)$$

The spills represent an extra level of generation capacity in the water system but which cannot be used in the energy mix at a particular time step due to other constraints and costs in the energy system. This excess capacity is distributed back to the hydropower plants and annually added to the existing required demand from plants based on the proportion of the deficit, given in Equation (2). This is a simplification that assumes a linear relationship between total country-level generation and generation from each hydropower plant in all time steps. In an actual case, where hydropower is highly nonlinear, this assumption in particular might be less efficient for a cascading hydropower system sharing the same river system. Then, this annual capacity is disaggregated to the monthly level based on the shadow prices. The idea behind this assumption is to give priority to months based on the relative value they will add to minimizing the cost while determining the energy mix. The shadow prices corresponding to maximum hydropower constraints in the energy model

are given by Equation (3). The reason for distributing the excess demand on an annual basis is the assumption that the reservoir storage exhibits annual cycles.

$$Demand\ Additional_{Annual,plant} = Spill_{Annual,country} * \frac{1 - deficit_{plant}}{\sum_c Annual\ Deficit_{Plant}} \quad (2)$$

$$\sum_l vProduction_{hsc,c,y,m,l,h} * pYearSplit_{l,m} \leq pHydroUpperLimit_{c,h,hsc,m} \quad (3)$$

The newly identified energy demands present a different water allocation scenario, which changes the water distribution and energy generation in the water system model and in extreme cases may affect generation in all plants. Therefore, the above step is executed iteratively to achieve equilibrium until either the total spill is zero or the amount of hydroelectricity utilized in the energy model converges to hydropower generation in the water system model.

The new established equilibrium does not necessarily represent an optimal configuration of hydropower generation for the water resource model. The distribution of spill based on shadow prices assures the optimal allocation of the spilled portion of demand; however, as the water model only explores the optimal allocation of resources for a time step, this does not guarantee optimal allocation over an entire year. However, given the limitation of the water model, a partial optimal point can be achieved that considers both the water availability and energy cost.

*2.3. Climate Scenarios*

The projections of changes in precipitation and temperature were derived from the hybrid frequency distributions (HFDs) of Schlosser et al. [96]. These are regionally downscaled model scenarios in the form of numerical hybridizations of 400 policy ensembles from the Massachusetts Institute of Technology (MIT) Integrated Global System Model (IGSM) [97,98] that correspond to 17 IPCC AR4 climate model results. The result is a meta-ensemble of climate change projections containing 6800 distinct members for possible adaptation. The MIT IGSM framework uses emission predictions and economic outputs from the MIT emission prediction and policy analysis model and earth system modeling predictions from the IGSM to drive a land system component, a crop model (CliCrop), and a water resource system model.

The HFD datasets characterize possible future climate outcomes, incorporating uncertainties in the structural differences of climate models, downscaling, and possible emission scenarios, as represented by different adopted policies. The two adopted policy scenarios corresponding to a restriction of global emissions of greenhouse gases to a concentration of 560 ppm $CO_2$ equivalent are referred to as level 1 stabilization (L1S) and unconstrained emissions. The scenario where no policy is adopted to constrain greenhouse gas emissions was considered in this assessment to compare the reduced level of impact as a result of policy adoption. The climate shocks were superimposed over historical reference case precipitation and temperature data obtained from the Climatic Research Unit (CRU) [99–102] to form climate scenarios.

Catchment runoff was simulated and hydrologic responses were characterized corresponding to the precipitation and temperature changes in each scenario using a conceptual hydrologic model called NAM (from the Danish Nedbør-Afstrømnings model). NAM is a deterministic, lumped, and conceptual rainfall–runoff model originally developed by the Institute of Hydrodynamics and Hydraulic Engineering at the Technical University of Denmark [103]. Climate change also affects irrigation water demand; thus, changes in the irrigation demand as a result of rising temperature were estimated for the scenarios.

## 3. Application of the Model to the Nile River Basin

We applied the integrated Water-Energy model to study the interactions between water and energy in the Nile River Basin, to assess the potential impact of climate change on the existing and future energy supply of countries within the basin, and to determine

how the electricity trade between countries might be affected by these changes over the period of 2010–2035, in monthly time intervals.

Hydropower is one of the principal sources of renewable energy in the Nile River Basin, where there is enormous opportunity for future development due to its highly underdeveloped nature. Climate-induced changes in the hydropower generating capacity will affect the existing and future dynamics of power trade among countries in the Nile River Basin. Although there is currently very limited cross-border electricity trading, several projects are in the pipeline, such as the Ethiopia Power Trade Project, the Nile Equatorial Lakes Interconnector Project, and the Regional Rusumo Falls Hydroelectric and Multipurpose Project (Figure 2); thus, hydropower generation will soon represent the dominant energy share in the basin, with increased levels of regional trade [104,105].

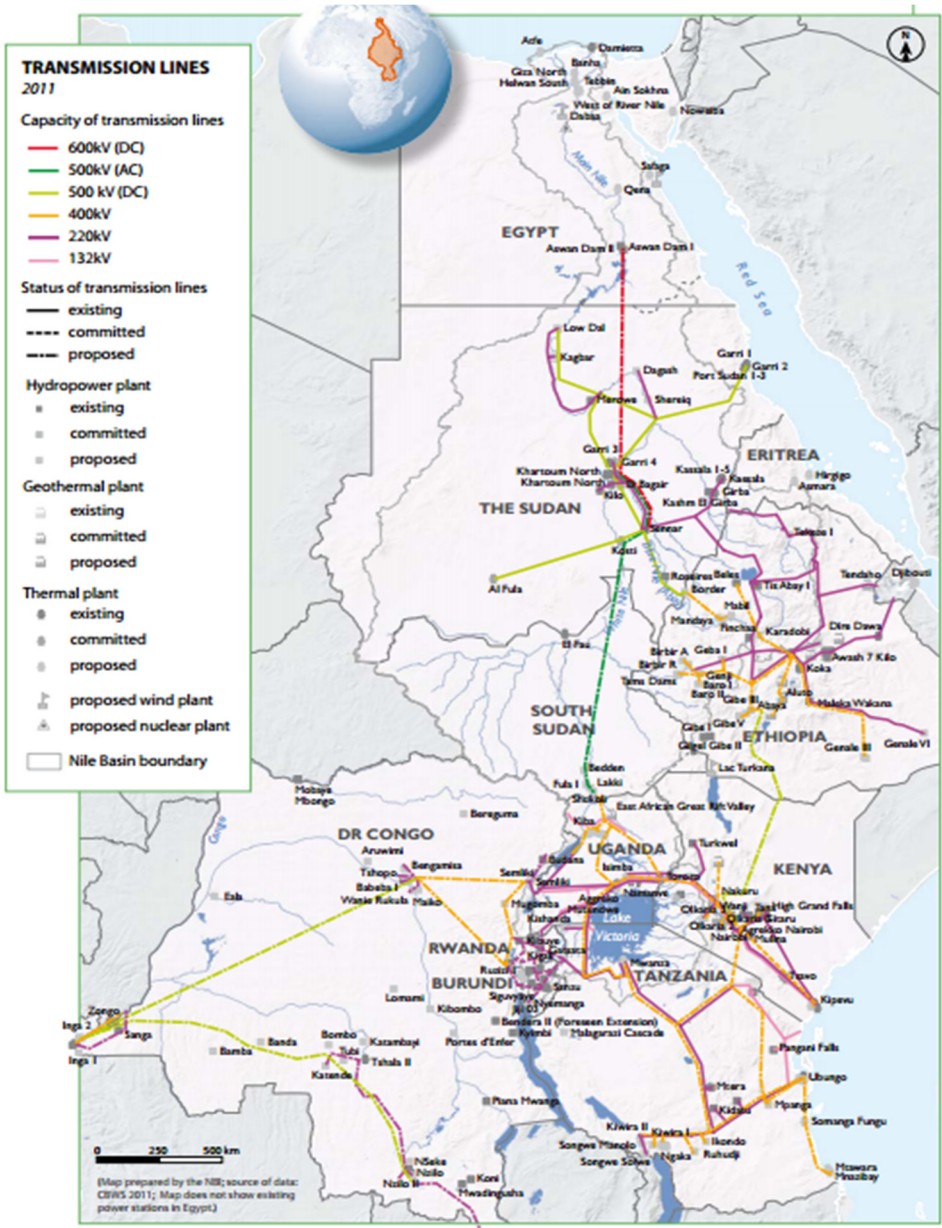

**Figure 2.** Planned and existing hydropower plants and regional interconnection links in the Nile River Basin (source: State of the Nile River Basin 2012, Nile Basin Initiative).

Hydropower generation was modeled on a monthly time step for existing and future hydropower facilities in the basin for a given level of initial estimates of the country-level

target demand. The energy demand for future plans was estimated as rough approximations obtained from either the project document or a percentage of the planned installed capacity of the plant. For run-of-river hydropower plants, the maximum generating capacity was taken as the power target for each month.

Hydropower plants that are located outside the Nile River Basin but contribute to country generation were considered to have a fixed generating capacity that did not vary with climate change. Although this was only an approximation used to simplify our analysis by limiting the hydrologic and water resource system analysis within the basin, the contribution from outside basins was proportionally small; thus, this simplification is unlikely to incur considerable error.

The existing capacity expansions and additional infrastructure expected to be built in the Nile River system were adopted from the PIDA development plan [91]. The level of capacity for all countries at five-year intervals is summarized [106,107] in Table 1. Djibouti is not within the Nile River system but it will be in the power grid; therefore, it was included in the energy model. The total installed capacity for the water model by the end of 2035 is approximately 24 TW, and the distribution is shown in Table 1. A study conducted by the Nile Basin Initiative (NBI) [108] indicated basin-wide power development options and trade opportunities. According to that study, the total energy demand in the Nile River Basin countries is expected to increase from 184 TWh in 2010 to 1170 TWh by 2035, representing an increase of 300% or more from current demand.

**Table 1.** Hydropower installed capacity for Nile River Basin countries (both within and outside the geographical extent of the Nile River Basin).

| Year | Burundi | Djibouti | Egypt | Ethiopia | Kenya | Rwanda | Sudan | Tanzania | Uganda |
|------|---------|----------|-------|----------|-------|--------|-------|----------|--------|
| 2010–2015 | 37 | 0 | 2250 | 1070 | 733 | 77 | 1727 | 561 | 830 |
| 2015–2020 | 103 | 0 | 2275 | 6182 | 733 | 148 | 1841 | 598 | 1660 |
| 2021–2025 | 103 | 0 | 2282 | 9677 | 733 | 225 | 2665 | 623 | 2444 |
| 2025–2030 | 103 | 0 | 2282 | 13,862 | 733 | 278 | 3301 | 623 | 2501 |
| 2030–2035 | 97 | 0 | 2282 | 13,862 | 733 | 278 | 3597 | 623 | 2501 |

## 4. Results

The results of hydropower contributions over the simulation period are shown for each country in Figure 3. We can see significant changes in hydropower penetration across different climate scenarios for Ethiopia, Egypt, and Sudan. For Burundi and Rwanda, hydropower generation comes from outside the basin; therefore, it is assumed to be fixed across climate scenarios. For the remaining countries, hydropower comes from run-of- river plants and generally increases as a result of the higher runoff expected in the majority of climate scenarios; however, the slight fluctuations caused by climate change are smoothed out in the energy model, indicating that all generated power will be used to the highest potential. The mix of energy results indicates that hydropower will continue to dominate the system power supply; examples for three selected countries are shown in Figure 4. The differences between observed and simulated value from 2015 to 2020 are negligible; the observed hydropower generation for each country resembled the corresponding simulated pattern as demonstrated in Figure 3.

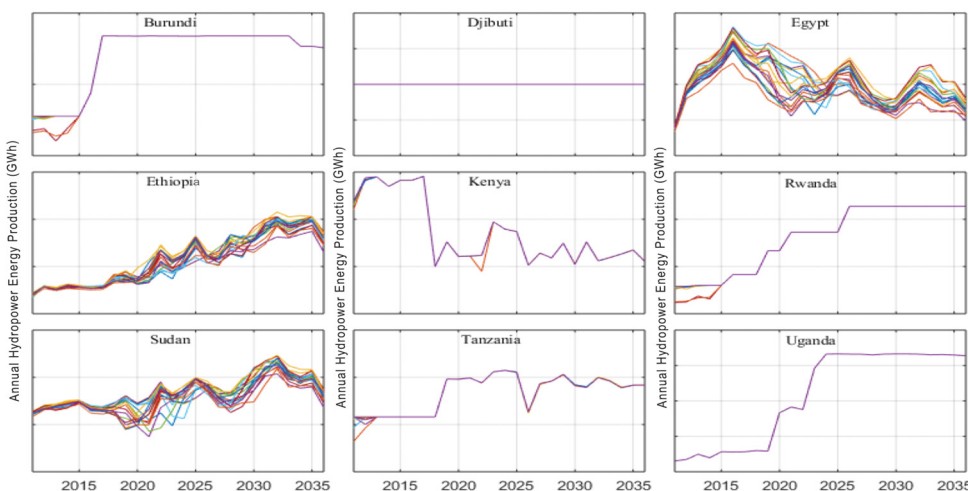

**Figure 3.** Annual hydropower energy production results Gigawatt hours (GWh) by country over the simulation period for selected climate change scenarios.

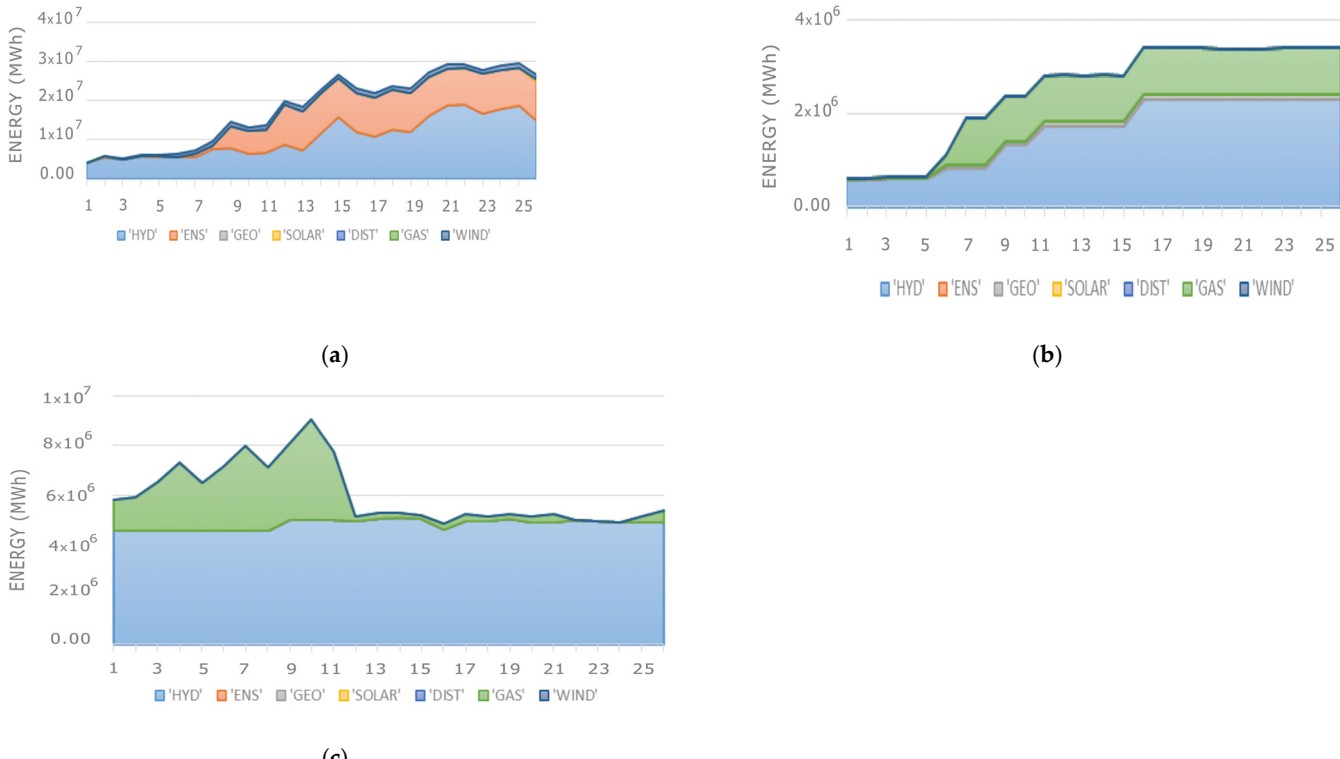

**Figure 4.** Energy mix trends over the simulation period for (**a**) Egypt, (**b**) Sudan, and (**c**) Tanzania.

The electricity trade results for the reference scenarios of the current (average for 2010–2014) and future (average for 2031–2035) conditions are shown in Figure 5. The results indicate that regional trade will grow stronger. Given the scenario assumption in the model, the highest degree of hydropower trade under conditions of no climate change is expected between Sudan and Egypt, and additional dams planned for the lower part of the Nile in Sudan will help meet Egypt's electricity demand and take preference over upstream dams (also see Figure 3).

|     | RWA | SUD | UGA | ETH | KEN | EGY | TAN | BUR |
| --- | --- | --- | --- | --- | --- | --- | --- | --- |
| BUR | 0.53 |     |     |     |     |     |     |     |
| EGY |     | 27.95 |     |     |     |     |     |     |
| KEN |     |     | 2.80 | 0.86 | 0.86 |     |     |     |
| RWA |     |     | 1.79 | 2.30 |     |     |     |     |
| SUD |     |     | 4.31 |     | 7.19 |     | 5.82 |     |
| TAN | 0.02 |     | 0.08 |     |     | 1.98 |     |     |
| UGA | 0.83 | 4.36 |     | 8.48 |     | 3.92 |     |     |
| ETH |     | 7.25 |     |     |     | 8.56 |     |     |
| DJI |     |     |     |     | 0.25 |     |     |     |

(**a**)

|     | RWA | SUD | UGA | ETH | KEN | EGY | TAN | BUR |
| --- | --- | --- | --- | --- | --- | --- | --- | --- |
| BUR | 0.15 |     |     |     |     |     |     |     |
| EGY |     | 1.97 |     |     |     |     |     |     |
| KEN |     |     | 0.98 |     |     |     |     |     |
| RWA |     |     | 0.45 |     |     |     |     | 0.01 |
| SUD |     |     | 1.00 |     | 2.46 |     | 1.98 |     |
| TAN | 0.00 |     | 1.78 |     |     |     |     |     |
| UGA |     | 1.43 |     | 0.09 |     | 0.44 |     |     |
| ETH |     | 1.56 |     |     |     |     |     |     |
| DJI |     |     |     |     |     | 0.02 |     |     |

(**b**)

**Figure 5.** Total energy trade in Terawatt hours (TWh) among countries for the (**a**) current (2010–2014) and (**b**) future (2031–2035) reference case scenarios with no climate change.

One notable result of this study is that climate change is predicted to alter trade relationships. In some climate change scenarios, trade dynamics exhibit a complete shift, whereas other scenarios lead to either a reduced or increased level of trade. The manner in which trade evolves under the selected 18 climate change scenarios is shown in Figure 6 for the five major regional trade relationships. Four of the 18 scenarios exhibit a change from the reference case with the highest level of trade shifting from Egypt–Sudan to Egypt–Ethiopia. Furthermore, in five of the 18 scenarios, trade between Ethiopia and Sudan will no longer have an economic advantage. Four of the 18 scenarios represent a stringent change in climate, which translates into a change in runoff. Under these conditions, Sudan may not build the planned dam adjacent to Egypt; instead, the Ethiopian highland may be preferable for large storage and dam construction due to its high altitude and low evaporation [109]. As a result, the electricity/energy-trading partnership is predicted to shift from Egypt–Sudan to Egypt–Ethiopia. In summary, trading relationships and economic gains will change under climate variability, and these must be considered to ensure an integrated and comprehensive regional response to mitigating the risk of future climate impacts.

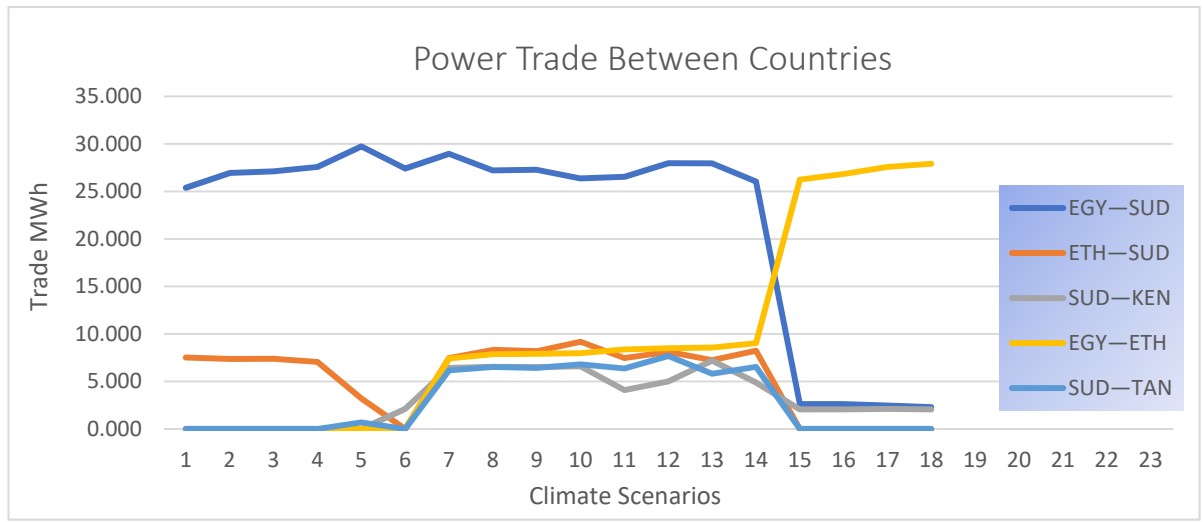

**Figure 6.** Average energy trade between countries for 2031–2035 under selected climate change scenarios (scenarios are sorted for better visualization).

## 5. Conclusions

This study proposed a framework for the integrated assessment of energy and water that captures the links among these sectors. This framework was applied to countries in the Nile River Basin to evaluate how energy will evolve under various climate change scenarios. The results indicated that, for selected scenarios, both the generating capacity and energy trade among countries could be significantly altered. Moreover, due to rapid economic growth in these countries, the demand for energy and water will continue to rise, stressing the shared river basin even more. Hydropower will continue to represent a major proportion of the energy supply in the Nile River Basin region; therefore, the integrated assessment of the energy and water nexus will be an essential component of planning for future development interventions and policy formulation.

The effects of climate change on the river system are critical and far-reaching. Countries where power generation depends heavily on hydroelectricity are highly vulnerable to climate variability. For example, the power generation capabilities of Egypt, Ethiopia, and Sudan fluctuate under climate variability; hence, their development is less sustainable. Moreover, energy trading plays a critical role in fostering cooperation and partnerships among the Nile basin countries. A change in climate could hamper trading relationships unless the countries develop a common framework to mitigate the adverse effects of climate change. A shift in trading partners and economic advantages from one country to another caused by climate change on shared water resources could also derail cooperation and lead to conflict and violence.

Water-Energy resources are an integrated part of sustainable development worldwide; however, growing climate variability on interannual to longer timescales will have an enormous effect on socioeconomic development, requiring adaptation on an individual, community, and national scale [110]. Energy is a crucial driving force for economic development as it fosters growth and transformation. Due to climate change uncertainties, governance of the Water-Energy system for greater economic benefits and regional cooperation becomes complicated and is compound by data limitations, which can impede robust analysis and assessment. The simple tailored model presented in this study provides a novel explanation for the Water-Energy governance strategy for a situation with limited data, thereby assisting policymakers and water resource planners in making decisions for the benefit of society and the economy. The proposed model is applicable to cases where the availability of complete and comprehensive data is inadequate.

A Water-Energy governance system that addresses climate variability and improved climate resilience are essential steps for reducing and managing future climate impacts. The proposed framework, which uses limited data through a tailored approach, can be used to undertake comprehensive and scientific analysis to inform policymakers about the dynamics of the Water-Energy governance system, including hydropower generation and allocation and future climate conditions to help water resource planners design and implement the necessary adaptations. Incorporating the findings of this assessment framework into both national and regional strategies can enhance the sustainable development of water and energy resources and foster trading partners and economic advantages. The Water-Energy analysis employed in this study can serve as a governance system for climate variability and risk management in cases with limited data to foster regional socioeconomic growth, increase resilience to climate vulnerability and variability, and enhance economic growth and cooperation at the policy and technical level.

Limitations were attributed to incomplete and inadequate data for verification and calibrations of the model to satisfy its minimum data requirements. The challenges were addressed through proxies and approximation techniques. The mode's flexibility is its core chrematistics of strength, and in terms of testing the model for a wide range of alternative scenarios is its limitation.

This analysis is carried out on foresight hydrological conditions. However, to account for different scales of variability, future studies should consider a wide range of synthetic hydrologic conditions that examine scenarios of annual, seasonal and decadal variability

over a more extended time to project the future climate of the Nile River basin and elucidate variations in water resources.

**Author Contributions:** Both authors contributed equally to the development of this article. A.Y. conceptualized methodological and theoretical formalism and performed the analysis. E.A. supervised the study. Both authors discussed the results and contributed to the final version of the manuscript. Both authors have read and agreed to the published version of the manuscript.

**Funding:** This research received no external funding.

**Data Availability Statement:** The data presented in this study are available on request from the corresponding author. The data are not publicly available due to legal and sensitivity reasons.

**Acknowledgments:** The authors would like to thank Yohannes Gebretsadik from Nile Basin Initiative (NBI), Professor Kenneth Marc Strzepek, Nardos Amdework, and other anonymous reviewers and the editor for their constructive comments, conversations, and feedback.

**Conflicts of Interest:** The authors declare no conflict of interest.

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
