# Peer review of "Assessment of the Water-Energy Nexus under Future Climate Change in the Nile River Basin"

_climate, doi:10.3390/cli9050084_

Round 1

Reviewer 1 Report

The paper brings some novelty and the scope is higly worthy of investigations. Authors shown the course of investigation clearly, results are supported by analyses. Paper is prepared properly. In my opinion paper could be published after minor revision:

  • carefully checked the paper (e.g. missing "e" in Energy in eq. 1; format of References);
  • adding a possibility of using the proposed model in other areas.

Summarized the paper is a good example of scientific article and I can recommend it to be published.

Author Response

Please kindly see attached our response. Thank you

Best regards,

Reviewer 2 Report

The article covers important topics related to the impact of climate change on the development of hydropower, taking into account the relationship between energy and water. The authors presented various scenarios of this modeling, taking into account the perspective of the Nile River.

The manuscript fits into the subject of the journal, however, it requires some corrections, which are described below. 1. Please provide the methodology with an addition to the limitations of the methods used.
2. I would recommend reorganizing the text as I get the impression that the aims of the article have been presented several times.
3. I would like to ask you to indicate what is the innovative nature of the manuscript, what new it brings to the development of knowledge within the scientific discipline.
4. What are the exact scenarios you have proposed? In the final charts I can see that there are a dozen of them, but I cannot see clearly what their variations are. I am asking for an answer and better clarification in the text.
5. In some places more references are missing, eg the second and third paragraphs of subchapter 1.1. If a certain knowledge is not universal and generally known, it should be referred to the relevant literature.
6. Please improve the quality Figure 2 - the map is slightly blurry.
7. I would recommend re-checking the text in terms of language and editing - there are, for example, typos or incorrect specialist terms (e.g. "runoff river plants" instead of "run-of-river hydropower plants" or "run-of-river hydroelectric power plants" ).
8. Please add a diagram illustrating the course of the tests performed.
9. The graphic materials presented should be corrected - e.g. in Figure 1 there is a black line on the left, for Figure 3 also axes with values ​​should be signed and a legend attached (it is not really known why Egypt, Ethiopia and Sudan have many lines of different colors, and other countries - in one color). You should remove the chart titles in the box (e.g. "Energy Mix trend for Egypt"). The captions under them serve this purpose. Please also remove all caps - e.g. "ENERGY (MWH)" in Figure 4 (should be "Energy (MWh)"). Additionally, I would recommend changing the number system - it should be in the power system, e.g. 4 · 104.
In Figure 9, please check that "000 MWH" is correct. Is it about 1000 MWh?
10. In Figure 5, the captions are the same on both sides. Additionally, Kenya is repeated several times in the first row - probably the countries should be analogous to those presented in the column.
11. Please check the citation method in the text - in some places you use the surname and the year at the same time, and then the numbers in square brackets. Additionally, in chapter three there was an error referring to a non-existent item in the bibliography.
12. Unfortunately, there are no conclusions. As it stands it is more of a summary. Please refer to the specific results.
13. The abstract needs improvement. It's very short, it doesn't show what the results of this article show, what results you've achieved.
14. The bibliography requires thorough editing in terms of editing - the text should be justified, check whether there are unnecessary spaces, spaces between quoted items, notation regarding the names of journals (there should be abbreviations and the entry itself in italics, and the year - in bold). The recording system itself does not meet the requirements for magazines published by MDPI. For unambiguous identification of the quoted items, please add DOI.

My main comments concern the organization of the text and editorial issues. Unfortunately, the article seems disordered and rough to the end. Please double-check the journal's text formatting requirements. Additionally, Sustainability is indicated in the header and footer, not Climate - please verify this.

Author Response

Please kindly find attached our response for your comments and thoughts. Thank you so much!

Best regards,

Reviewer 3 Report

The current study investigates the short- and long-term impact of climate change on the water-energy nexus and resources to capture their interlinkage for the estimation of hydropower generation constrained by water availability in several countries along the river Nile. The general idea of this work is interesting. Please see some major issues to be addressed:

1) In the Introduction, long-term several models for the simulation of water-energy nexus are mentioned, with most of them being deterministic. However, for the simulation of the long-term uncertainty of the natural processes included in the water-energy nexus, one should perform a stochastic rather a deterministic analysis, in order to be able to capture the variability of the system. Although the PRIMES model is mentioned as stochastic, it is not exactly that. In fact, it is mentioned that "The stochastic or variable RES (wind, solar PV, solar thermal, small hydro, tidal-wave) are represented as a deterministic equivalent power capacity:" (E3Mlab, 2008). Purely stochastic models for the water-energy nexus systems have been only recently developed (see such examples: a literature review on stochastic optimization on water-energy nexus in Vakilifard et al., 2018; a literature review the water-energy assessment methods including the effect of food forming the water-energy-food nexus (which are mostly always connected) in Albrecht et al., 2018; a literature review on water-energy stochastic simulation in Mamassis et al., 2021).

2) Please consider giving more mathematical expressions and details on the energy system model (RIPA). For example, how does RIPA estimated the energy exchange rate among countries, the energy cost, and demand in each country, the energy-water connection etc.

3) In the text, it is mentioned that for the simulation of the Water-related processes runoff, temperature, and precipitation are modeled. However, I could not find many details in the text on the mathematical description of their model. Please consider giving more details on the model. Also, please discuss whether the model used includes important properties such as preservation of the long-term persistence (LTP) of these processes, which is considered vital for this work that aims to simulate the long-term impacts on the water-energy system. The LTP behavior has been recently shown (Dimitriadis et al., 2021) to exist in these processes (i.e., streamflow, temperature, precipitation) based on a global-scale analysis in thousand of ground stations (including areas at the river Nile). The LTP is important in the current study since it describes the increased uncertainty of the water-energy-nexus related processes, which also affect the water-energy system's uncertainty.

4) Figure 3:

4a) Figure 3 is very interesting and illustrates how the uncertainty is expected to increase through the future years due to the changing climate and the LTP behavior (explained in the previous comment). Please consider showing in Figure 3 how the standard deviation increases in each simulation year, and discuss which country is expected to have the highest and lowest variability. These two countries will have the highest and lowest, respectively, effect on the water-energy system management.

4b) Also in Figure 3, please explain why some countries have only one simulation.

4c) Also in Figure 3, is it possible to show the observed values of Annual hydropower energy production from 2015 to nowadays for comparison to the simulated ones?

5) Please considering giving more details on the application on how one determines which energy is related to water and which to other resources. Also, does the current analysis includes any water-energy losses, or does it cover water needs for other uses (such as agricultural, domestic, urban etc.)?

6) Please give more information on how the Results are supported by the Analysis and the Application. For example:

6a) It is mentioned in the Conclusions that in the current study a Framework is illustrated in which the interlinkage of energy and water is captured. To support such a result more details should be included in the text on how this linkage is mathematically simulated, what are the limitations and advantages of the presented framework etc.

6b) It is mentioned in the Conclusions that the presented framework was applied to the countries along the Nile River Basin to evaluate how energy will evolve in the future under climate change. However, the energy forecast is based on the water-status of each country (e.g., in the hydropower dams). Have you considered simulating these levels? Also, how Water is included in the Application at the river Nile, is it through the climate change scenarios?

7) Please perform a strong grammar and syntax check throughout the text, since I have tracked several errors. For example:

7a) "our model has developed to captures the unique..." should be "our model has been developed to captures the unique...".

7b) "It has long been understood that there is a strong interlinkage between water resource systems and energy systems." could be "It has been long realized that there is a strong interlinkage between water resource systems and energy systems".

7c) "The level of capacity for the countries at five-year intervals is summarized in Error! Reference source not found.". Please check all References in the text.

References

Albrecht, T.R., A. Crootof, and C.A. Scott, The water-energy-food nexus: A systematic review of methods for nexus assessment. Environ. Res. Lett., 13, 43002, 2018.

Dimitriadis, P., D. Koutsoyiannis, T. Iliopoulou, and P. Papanicolaou, A global-scale investigation of stochastic similarities in marginal distribution and dependence structure of key hydrological-cycle processes, Hydrology, 8 (2), 59, doi:10.3390/hydrology8020059, 2021.

E3Mlab, PRIMES model, 2008 [http://www.e3mlab.ntua.gr/manuals/The_PRIMES_MODEL_2008.pdf].

Mamassis, N., A. Efstratiadis, P. Dimitriadis, T. Iliopoulou, R. Ioannidis, and D. Koutsoyiannis, Water and Energy, Handbook of Water Resources Management: Discourses, Concepts and Examples, edited by J.J. Bogardi, T. Tingsanchali, K.D.W. Nandalal, J. Gupta, L. Salamé, R.R.P. van Nooijen, A.G. Kolechkina, N. Kumar, and A. Bhaduri, Chapter 20, 617–655, doi:10.1007/978-3-030-60147-8_20, Springer Nature, Switzerland, 2021.

Vakilifard, N.; Anda, M.; Bahri, P.A.; Ho, G. The role of water-energy nexus in optimising water supply systems—Review of techniques and approaches.Renew. Sustain. Energy Rev., 82, 1424–1432, 2018.

Author Response

Please kindly see attached our response. Thankyou. 

Round 2

Reviewer 2 Report

  Thank you for responding to my comments. The article is of higher quality in its current form.

However, I would suggest improving the following points:
- Please prepare the main text as continuous, with no spaces between paragraphs (e.g. as at the beginning of page 3).
- Figure 3 - the values in each graph should be marked on the Y axis (I suppose they are different in each graph).
- Figure 4 - I would avoid marking values as "x.xxE + xx".
- Figure 5b - it should be "Future case scenario average 2031-2035".
- Figure 6 - it should be "Trade (MWh)", please remove the title "Power Trade Between Countries" in the middle of the chart.
- Abbreviations should be avoided in the bibliography; in addition to them, please provide the full names of the parties/organizations/institutions.
- Please, standardize the styles in the bibliography - the font and size should be the same, justified, line spacing 1, added DOI number.

I hope you can publish this article. I wish you all the best.

Author Response

Thank you for your constructive review; your thoughts, comments, insights, and feedback have greatly improved the quality of the manuscript. We have addressed your additional comments and recommendations in the attached document. We appreciated your thoughts and insights. Thank you.

With best regards,

Reviewer 3 Report

The manuscript has been improved, however, there are still 3 major issues that need to be addressed:

1) Since the focus of this study is the long-term management of water-energy nexus, and since it includes long-term models for the river-flow, temperature, and rainfall, then the presence of the Long-Term Persistence (LTP) in all these natural processes should be discussed (as recommended in the 3rd comment of my previous review). The LTP is not just a technical issue, since it corresponds to the quantification of the high variability of those processes and ultimately to the climatic degree of variability and change, which is one of the basic themes of this paper.

2) In Figure 3, there should be a comparison (or at least a discussion) on the observed vs. the simulated values of annual hydropower energy production from 2015 to nowadays for comparison and validation of the model (as recommended in the 4c comment of my previous review), which is one of the basic themes of this paper.

3) Please perform a strong grammar and syntax check, since there are still multiple such issues in the text (as recommended in the 7th comment of my previous review).

Author Response

Thank you for the constructive review; your comments, thoughts, and recommendations have greatly improved the quality of the manuscript. We have addressed your suggestions and recommendations as follows in the attached document. We appreciated your thoughts and insights! Thank you.

With best regards,

This manuscript is a resubmission of an earlier submission. The following is a list of the peer review reports and author responses from that submission.